# *Rhipicephalus microplus* and Its Impact on *Anaplasma marginale* Multistrain Infections in Contrasting Epidemiological Contexts

**DOI:** 10.3390/pathogens14020160

**Published:** 2025-02-07

**Authors:** Agustina E. Pérez, Eliana C. Guillemi, Nestor F. Sarmiento, Germán J. Cantón, Marisa D. Farber

**Affiliations:** 1Instituto de Agrobiotecnología y Biología Molecular (IABIMO), INTA-CONICET, Hurlingham B1686LQF, Argentina; perez.agustina@inta.gob.ar (A.E.P.); farber.marisa@inta.gob.ar (M.D.F.); 2Estación Experimental Agropecuaria Mercedes, Instituto Nacional de Tecnología Agropecuaria, Mercedes 3470, Argentina; sarmiento.nestor@inta.gob.ar; 3Instituto de Innovación para la Producción Agropecuaria y el Desarrollo Sostenible (IPADS), INTA Balcarce-CONICET, Balcarce 7620, Argentina; canton.german@inta.gob.ar

**Keywords:** *Anaplasma marginale*, *msp1α*, epidemiology, genotypes, tick borne, complex infections

## Abstract

Bovine anaplasmosis is a disease caused by *Anaplasma marginale*, a tick-borne bacterial pathogen with global distribution, primarily determined by the range of its vector. In Argentina, *Rhipicephalus microplus* is the main species associated with *A. marginale* transmission, even though this bacterium can also be mechanically transmitted. We studied complex infections (more than one *A. marginale* variant) in naturally infected bovines from two different epidemiological contexts: a region with the tick vector and a tick-free region. In the tick-free area, symptomatic infections were associated with a single *A. marginale* genotype, while asymptomatic bovines from the same herd remained chronically infected with a low number of genotype variants. By contrast, in the region where *R. microplus* is present, the only symptomatic bovine showed highly diverse infections, with 19 distinctive genotypes. Additionally, *A. marginale* genotypes were also detected in tick tissues. These findings, together with previous data, indicate that *R. microplus* harbors *A. marginale* populations that are maintained through tick generations by means of transovarial transmission. Furthermore, this tick species is responsible for maintaining *A. marginale* diversity in the bovine host over time through coinfection and superinfection events.

## 1. Introduction

Bovine anaplasmosis is an important tick-borne disease affecting cattle and other mammals, with a worldwide distribution [1,2,3,4,5]. The disease is caused by *Anaplasma marginale*, a Gram-negative alpha-proteobacteria from the *Anaplasmataceae* family in the order Rickettsiales. This obligate intracellular microorganism pathogen infects erythrocytes, thus leading to clinical signs such as fever, lethargy, icterus, reduced weight gain and, in severe cases, death [6]. Infected cattle that survive acute anaplasmosis may also develop persistent infections characterized by low levels of rickettsemia. These persistently infected (carrier) animals may act as important reservoirs of infection, contributing to the disease’s transmission dynamics [6,7]. Given the central role that the beef and dairy industries play in Argentina’s economy, with significant social and economic implications, understanding the epidemiology of bovine anaplasmosis and its transmission mechanisms is essential for effective disease control and prevention strategies.

*A. marginale* can be transmitted by hematophagous arthropods or iatrogenically, with the latter often linked to poor management practices such as the use of contaminated medical instruments [6]. Among arthropods, ticks are the predominant vectors worldwide. In Argentina, the main vector species is *R. microplus* [8]. Owing to Argentina’s diverse environmental and climatic conditions, *R. microplus* is prevalent in regions with optimal humidity and temperature, such as northern Argentina, but it is absent from other regions (central and southern areas of Argentina) because of their suboptimal conditions for its development [8,9,10]. In tick-free regions, the dispersion of *A. marginale* is associated with the translocation of infected cattle from anaplasmosis-endemic areas [6]. Recently, molecular evidence has also suggested the role of horseflies in the transmission of *A. marginale* [11].

The use of molecular markers is a valuable tool for studying disease epidemiology. Regarding anaplasmosis, the molecular epidemiology of *A. marginale* can be assessed by amplifying and sequencing a fragment of the *msp1α* gene, which encodes the MSP1a protein. The MSP1a sequence varies in size among isolates due to differences in the number of tandemly repeated 28–29 aminoacid peptides [12]. Complex infections occur when multiple strains of *A. marginale* are coinfecting a host. This condition results from the acquisition of two or more variants and can correspond to a superinfection or a coinfection. The first occurs when multiple strains are transmitted independently in separate infective events, while coinfection takes place when more than one variant is transmitted in a single infective event [13,14]. In Argentina, molecular characterization of *A. marginale* strains has been limited to endemic regions where *R. microplus* is present [8,15,16].

In the present study, we aimed to analyze *A. marginale* strains in three infected bovine hosts and investigate their transmission dynamics in two different epidemiological contexts: with and without the presence of *R. microplus*. In each scenario, we assessed the characteristics of *A. marginale* MSP1a variant complex infections.

## 2. Materials and Methods

### 2.1. Sample Collection and Processing

This study was conducted under the guidelines of the Institutional Committee for the Use and Care of Experimentation Animals (CICUAE), Corrientes, Argentina (protocol numbers 02/2018 and 04/2022).

The hosts and samples, as well as the molecular tools employed in this study, are represented schematically in Figure 1. Briefly, *A. marginale*-infected cattle from two different geographic areas were studied: a tick-free area (Buenos Aires province, Argentina) and a region where *R. microplus* is widely distributed (Corrientes province, Argentina) [10]. In the tick-free area, both clinically symptomatic cattle and asymptomatic bovines from the same herd were studied. In Corrientes province, a clinically symptomatic bovine and the ticks parasitizing it were investigated.

Buenos Aires province (tick-free area): An outbreak of anaplasmosis was registered in May 2019 on a beef cattle farm located in Lincoln department (35°05′49″ S 61°50′16″ W), affecting 7 out of 45 18-month-old Aberdeen Angus bulls. These bulls were born in the Cordoba and Entre Rios provinces, regions known to be endemic for anaplasmosis and probable tick infestations [10]. Six months after arrival, one of the affected bulls (bovine BA2019) died after presenting acute clinical manifestations (fever, lethargy and severe anemia). Necropsy revealed gross (icterus and splenomegaly) and histological lesions (Figure 2). The spleen was sampled and preserved at −20 °C until further analysis. Two years later, in 2021, blood samples (10 mL) were collected from the jugular vein of 20 Aberdeen Angus bulls from the same herd affected in 2019. The samples were collected and preserved in 3.8% sodium citrate at −20 °C for further analysis. One of the 15/20 PCR-positive samples was selected for *A. marginale* characterization (bovine BA2021).

Corrientes province (tick presence region): In July 2021, at a beef cattle farm located in Monte Caseros (30°14′21″ S 57°57′34″ W), a 3-year-old Braford cow developed acute anemia, weakness and fever (bovine C2021). Seven blood samples (10 mL) were collected from animals of the same herd, which also exhibited a high tick burden of *R. microplus*. Ten female adult ticks were collected from each affected cow and kept in a plastic tube with 70% ethanol for further organ dissection.

DNA from bovine blood samples was extracted from 400 μL of blood using the ADN PuriPrep-S kit (INBIO Highway, Buenos Aires, Argentina) according to the manufacturer’s instructions and kept stored at −20 °C until further use. DNA from the spleen sample was extracted with the Nucleospin Tissue Kit (Macherey-Nagel, GmbH & Co. KG. Düren, Germany) following the manufacturer’s instructions and stored at −20 °C until further use. Tick samples preserved in 70% ethanol (Biopack, Buenos Aires, Argentina) were processed. The ticks were identified as *R. microplus* using taxonomic keys before dissection [17]. The arthropod surface was disinfected with 70% ethanol (Biopack, Buenos Aires, Argentina) for 60 s, rinsed with sterile PBS (Sigma-Aldrich, Saint Louis, MO, USA) and air-dried. Dissection was performed in a 5 cm sterile Petri dish under a stereoscopic magnifier (SMZ-2T Nikon, Sendai, Natori, Japan). Ticks were stabilized with double-sided tape and rinsed in 1 mL of sterile PBS. An incision was made with a scalpel blade to release the dorsal cuticle and the organs were removed using sterile forceps and needles [18,19]. Ovaries and salivary glands were identified, extracted, washed with sterile PBS and placed individually in 1.5 mL tubes containing 30 μL of sterile PBS. DNA from ovaries and salivary glands was extracted using the Nucleospin Tissue Kit following the manufacturer’s instructions and stored at −20 °C until further use.

### 2.2. Pathogen Detection

*Anaplasma marginale* was identified by amplifying a fragment of *msp1β*, a three-copy gene that encodes the outer major surface protein MSP1b [20]. Negative samples were tested to verify DNA extraction quality: bovine samples were targeted for the mitochondrial *16s rRNA* gene common to mammals [21]; tick samples were targeted for the tick’s *16s rRNA* gene [22].

Molecular amplifications were performed on a T21 thermal cycler (IVEMA, Buenos Aires, Argentina) using a 20 μL reaction mixture containing 0.4 µM of each primer (Operon Eurofins, Louisville, KY, USA), 200 µM of each deoxyribonucleotide triphosphate (INBIO Highway, Buenos Aires, Argentina), 0.02 U/µL of Q5 High-Fidelity DNA Polymerase (NEB, Whitby, ON, Canada), 1 PCR buffer (NEB, Whitby, ON, Canada), 1X PCR Enhancer (NEB, Whitby, ON, Canada) and ~100 ng of genomic DNA. Positive (DNA from *A. marginale* Mercedes reference strain) and negative (MiliQ^®^ water, Direct-Q 5 UV, Millipore, Merk, Burlington, MA, USA) controls were included in the assay. An aliquot of each amplified product (15 μL) was analyzed by electrophoresis in 1.5% agarose gel (INBIO Highway, Buenos Aires, Argentina) stained with ethidium bromide (INBIO Highway, Buenos Aires, Argentina) and a molecular size marker (1 Kb Plus DNA Ladder, NEB, Whitby, ON, Canada).

Following *A. marginale* molecular detection, blood DNA samples were quantified by pPCR, and bovine and tick samples were characterized by amplifying, cloning and sequencing the *msp1α* gene (Figure 1).

### 2.3. Molecular Quantification

Positive samples to *msp1β* were quantified by qPCR as described by Carelli, 2007 [23]. A standard curve was constructed by cloning the 95 bp *A. marginale* PCR amplicon into a pGEM-T easy^®^ vector (Promega, Madison, WI, USA). Recombinant plasmids were purified using a Plasmid Miniprep Kit (Norgen, Biotek Corp. Thorold, ON, Canada) and prepared to perform ten-fold dilutions with MiliQ^®^ water ranging from 1 ng/μL (5,134,000 copies) to 1 × 10^−5^ ng/μL (51.34 copies). All qPCR amplifications were performed in triplicate on a StepOnePlus™ Real-Time PCR System (Thermo Fisher Scientific, Waltham, MA, USA) using a 20 μL reaction mixture containing 1 of iTaq universal probes supermix (BioRad, Hercules, CA, USA), 0.4 µM of each primer (Operon Eurofins, Louisville, KY, USA), 0.3 µM of probe (Operon Eurofins, Louisville, KY, USA) and ~150 ng of genomic DNA sample.

### 2.4. Molecular Characterization

*A. marginale* genotypes present in the different samples were assessed by amplifying and sequencing a fragment of the *msp1α* gene [24]. Molecular amplifications were performed on a T21 thermal cycler (IVEMA, Buenos Aires, Argentina) in a 50 μL reaction mixture containing 0.4 µM of each primer (Operon Eurofins, Louisville, KY, USA), 200 µM of each deoxyribonucleotide triphosphate (INBIO Highway, Buenos Aires, Argentina), 0.02 U/µL of Q5 High-Fidelity DNA Polymerase (NEB, Whitby, ON, Canada), PCR buffer (NEB, Whitby, ON, Canada), PCR Enhancer (NEB, Whitby, ON, Canada) and ~200 ng of genomic DNA. An aliquot of each amplified product (15 μL) was analyzed by electrophoresis in 2% agarose gel (INBIO Highway, Buenos Aires, Argentina) stained with ethidium bromide (INBIO Highway, Buenos Aires, Argentina) and a molecular size marker (1 kb DNA ladder, NEB, Whitby, ON, Canada).

In cases where more than one *msp1α* fragment was visualized, amplicons were cloned and sequenced to detect the coexisting genotypes. The amplified products were purified using a commercial kit (Monarch Spin PCR & DNA Cleanup Kit, NEB, Whitby, ON, Canada) according to the manufacturer’s instructions. The purified PCR fragments were cloned into the pGEM-T easy^®^ vector (Promega, Madison, WI, USA), following the manufacturer’s instructions, and transformed into DH5α *Escherichia coli* competent cells (prepared in-house). The selection of recombinants was made on LB/ampicillin plates to which an IPTG/X-Gal solution was applied (prepared in-house). Recombinant plasmids from white colonies were purified using the Plasmid Miniprep Kit (Norgen, Biotek Corp. Thorold, ON, Canadá) and sequenced using the universal primers T7 and SP6 with a BigDye Terminator v3.1 kit. The sequences were analyzed on an ABI 3500 genetic analyzer (Applied Biosystems, Woburn, MA, USA), at the Genomic Unit (IABIMO, INTA-CONICET, Buenos Aires, Argentina). Both strands of the plasmid were sequenced for greater reliability. The number of analyzed recombinant clones was 4 for the BA2021 sample, 32 for the C2021 sample and 8 for each tick sample (salivary glands (C2021_SG) and ovaries (C2021_O)). The complementary nucleotide sequences of each fragment were assembled using the Vector NTI Advanced 10 program (Invitrogen, Waltham, MA, USA). Translation into the MSP1a protein was performed using the translate tool from the Expasy website [25]. Tandem repeats were manually identified using the RepeatAnalyzer updated database [26].

## 3. Results

### 3.1. Pathogen Detection and Molecular Quantification

By PCR amplification of the *msp1β* gene, we confirmed *Anaplasma marginale* infection in BA2019, in 15 of the 20 bovine blood samples from 2021 (including BA2021) and in 1 of the 7 blood samples from Corrientes province (C2021). Quantitative real-time PCR revealed that symptomatic animals (BA2019 and C2021) exhibited higher bacterial load than asymptomatic animals (BA2021 and the rest of the positive samples quantified) (Table 1). Additionally, *A. marginale* DNA was detectable in the ovaries (C2021_O) and salivary glands (C2021_SG) of *R. microplus* collected from C2021.

### 3.2. Strain Genotyping Through the Variable Region of the msp1a Gene

Amplification of the *msp1α* gene from the BA2019 sample yielded a single amplification fragment in the agarose gel, and its sequencing, along with its translated amino acid sequence, revealed a single genotype (Figure 3A). On the other hand, the analysis of BA2021 resulted in two amplification fragments of different sizes. Sequencing of the four recombinant clones analyzed resulted in two genotypes (different from the one detected in BA2019) (Figure 3A).

Amplification of the *msp1α* gene from sample C2021 produced seven PCR products. Analysis of the 32 recombinant clones sequenced revealed 19 different genotypes (Figure 3B). The most frequently identified genotype was AR9 62 62 62 61, found in 9 out of 32 clones. In tick organs from C2021, two amplification fragments of different sizes were detectable in C2021_SG and C2021_O. Sequencing identified three genotypes in C2021_SG, with AR9 62 62 62 61 being the most frequently detected (in six of eight clones). In C2021_O, two genotypes were detectable, and, as for C2021_SG, the variant AR9 62 62 62 61 was the most frequently detected (in seven of eight clones) (Figure 3B and Figure 4).

Figure 5 shows the alignment of the MSP1a amino acid repeat sequences of all *A. marginale* genotypes detected in this study.

The nine novel repeats identified in the present study have been designated AR9 through AR17. Sequence variants corresponding to these genotypes have been submitted to GenBank (accession numbers: PQ821006-PQ821010 and PQ873055-PQ873102).

## 4. Discussion

In the present study, we analyzed the molecular diversity of *A. marginale* infections in bovines from two different ecological settings in Argentina. The analyzed samples came from a symptomatic and an asymptomatic naturally infected bovine from Buenos Aires province, which is an *R. microplus*-free area, and from a symptomatic bovine from Corrientes province, where this tick species is present. While the symptomatic bovine from the tick-free area (Buenos Aires) was infected by a single *A. marginale* genotype, the symptomatic bovine from the *R. microplus*-endemic region (Corrientes province) showed a complex infection consisting of 19 different genotypes. Researchers have previously reported this high degree of genotype diversity in *A. marginale* infections in Argentina [15,16] and other tick-endemic areas [15,27,28]. However, to the best of our knowledge, no studies have addressed the genotypic diversity of *A. marginale* complex infections in tick-free areas or discussed the factors driving these patterns.

BA2019 was part of a herd transported from Córdoba province in 2019. Six months after arriving in Buenos Aires province, BA2019 presented clinical signs of anaplasmosis, thus prompting treatment with oxytetracycline for both this animal and the entire herd. Although BA2021, a member of that herd, did not exhibit clinical signs, it was also treated preventively. The detection of *A. marginale* variants in BA2021 samples and their quantification two years after the translocation event indicates that the coinfection (comprising two genotypes transmitted in a single infective event) that occurred in the original location (Córdoba province) persisted in low bacteriemia loads over two years in this bovine. The absence of the tick vector in the destination province (Buenos Aires) prevented the occurrence of further superinfection events [14].

Regarding all the other asymptomatic bovines from the herd that were translocated to Buenos Aires, similar to BA2021, qPCR testing yielded low copy numbers in 15 of the 20 positive animals. In a previous report studying *A. marginale*’s resistance to long-acting oxytetracycline in bovines, the researchers concluded that some genotypes might be resistant to the drug, thus leading to treatment failure probably due to genotype sensitivity variations, among other factors [15]. While the current study did not test the resistance of *A. marginale* variants to oxytetracycline, it is possible that those genotypes detected in bovines from Buenos Aires after treatment may have exhibited higher resistance profiles. Even if such selection occurred, no further superinfection events were possible since the tick vector is not present in this area [10].

Bovine C2021, from the tick-endemic region, showed a highly diverse profile consisting of 19 genotypes, while tick samples associated with this bovine showed lower diversity (three genotypes in C2021_SG and two in C2021_O). Despite this, a particular genotype (AR9 62 62 62 61) was the most prevalent among all tested samples for this symptomatic case, including bovine blood, tick ovaries and tick salivary glands. Additionally, this common genotype contained a previously unreported repeat (AR9). These findings suggest that the tick not only serves as a vector but may also play a key role in maintaining *A. marginale* diversity across space and time through coinfection and superinfection events. The role of *R. microplus* in sustaining complex infections is likely due to the transovarial transmission of *A. marginale* to tick progeny, which enables unfed larvae to harbor and transmit the pathogen to a new bovine host [16]. Furthermore, the movement of adult ticks among cattle could contribute to shaping genotypic diversity through new, independent infection events [6,14].

In this study, we analyzed the population structure of *A. marginale* variants in endemic and non-endemic areas using the specific molecular marker MSP1a and identified two kinds of *A. marginale* populations associated with the mammalian host (bovine) and the arthropod vector (tick). The degree of diversity and the genotype profile found in both ticks and bovines depend on specific host–pathogen and vector–pathogen interactions. Among these factors, we can mention the microbiome of both bovines and ticks, the immune systems of mammals and arthropods and the antigenic surface proteins that interact with those immune systems, among other factors [24,29]. As a result, *A. marginale* strains show different fitness profiles, thus leading to the emergence of different genotype profiles. Ticks harbor *A. marginale* populations that are maintained through tick generations via transovarial transmission and modified by the ingestion of new *A. marginale* variants during blood meals [16]. Moreover, ticks are responsible for transmitting *A. marginale* to the mammalian host, therefore serving as a source of new variants through both coinfection and superinfection events. These new genotypes, transmitted by ticks to bovines, contribute to increasing the *A. marginale* population diversity in this vertebrate host.

The animals from the Buenos Aires herd were originally from Córdoba, a province where *R. microplus* is present in some areas, primary in the north [10]. These animals were around eight to twelve months old when moved to Buenos Aires and, at that moment, were not parasitized by ticks nor showing any clinical signs of disease (both are requirements for moving cattle from a tick-endemic area to a tick-free area in Argentina). During the anaplasmosis outbreak (BA2019), no iatrogenic practices could be associated with the disease occurrence in the preceding weeks, and all the herd must have been infected when moved from Córdoba. However, as young cattle are resistant to the disease, the animals did not show clinical signs until they were older [7]. Since *R. microplus* is absent in Buenos Aires, no further tick transmission events occurred, and the reduced *A. marginale* populations in the bovine host were modulated by pathogen–host interactions [29]. This explains why asymptomatic bovines were *A. marginale*-positive and harbored a low number of genotypes two years later, in 2021. Regarding *A. marginale* epidemiology in tick-free areas, we should mention that other hematophagous insects, such as tabanids, have also been associated with mechanical transmission [11].

Understanding the role of *R. microplus* in the complex infection dynamics of *A. marginale* is the next step towards a deeper understanding of the ecology of anaplasmosis [13]. The reduced sample size analyzed in this study is a limitation, and further research with a larger number of samples will contribute significantly to a better understanding of the epidemiology of anaplasmosis. This study represents the first analysis of *A. marginale* complex infections in a tick-free area and provides insight into the factors driving these patterns of infection. Additionally, new evidence is presented regarding the transovarial transmission of *A. marginale* in the *R. microplus* tick vector and its relevant role in maintaining complex infections in the bovine host.

## 5. Conclusions

In this study, we analyzed *A. marginale* complex infections in naturally infected bovines from two areas in Argentina. Our findings suggest that the presence of *R. microplus* shapes the highly diverse genotype profiles found in bovines within tick-endemic areas. By contrast, bovines in the tick-free area carried a low-diversity genotype profile. To the best of our knowledge, this is the first study analyzing *A. marginale* complex infections in a tick-free area. Further studies involving larger sample sized will be essential for advancing our understanding of *A. marginale* complex infections in both tick-endemic and tick-free regions.

## Figures and Tables

**Figure 1 pathogens-14-00160-f001:**
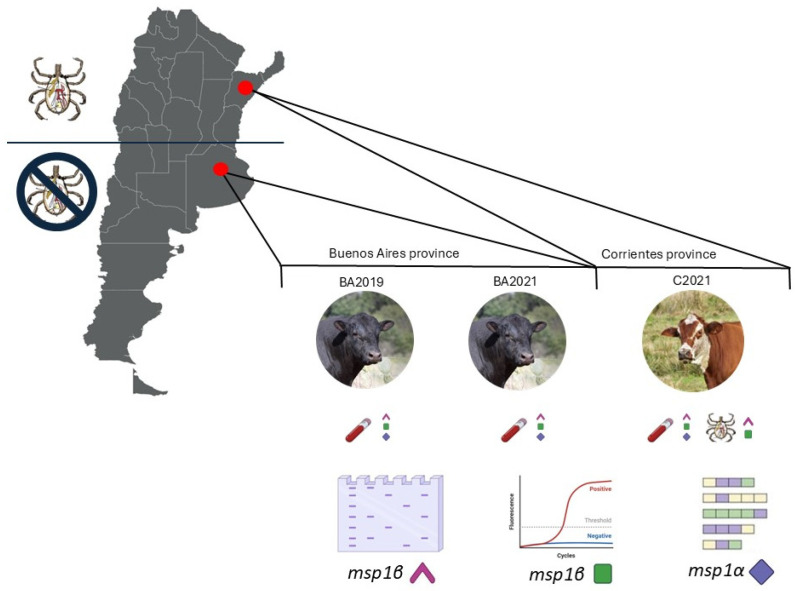
Schematic representation of the study design. Bovines from two regions in Argentina were studied. Two bovines from Buenos Aires province were sampled (spleen from the symptomatic case in 2019 and blood from the asymptomatic bovine in 2021) and one bovine from Corrientes province (blood and ticks from a symptomatic case in 2021). All samples were analyzed through a diagnostic end-point PCR (*msp1β*) and a molecular characterization based on sequencing a tandemly repeated genomic region (*msp1α*). For bovine samples, we also performed a qPCR for molecular quantification (*msp1β*).

**Figure 2 pathogens-14-00160-f002:**
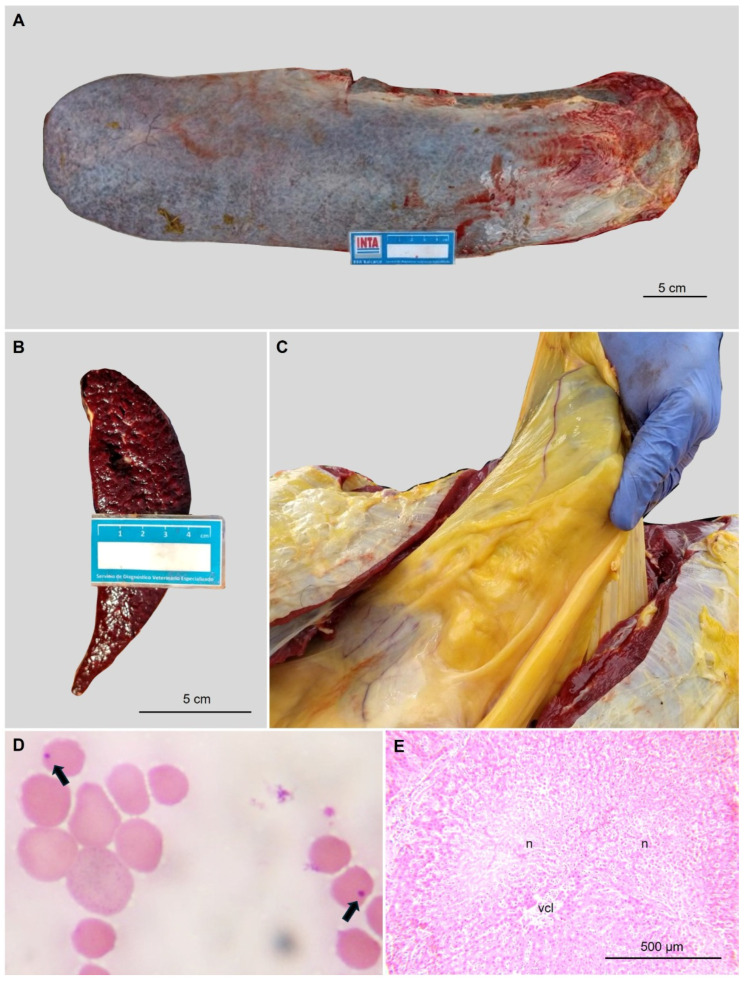
Macroscopic and histological lesions observed in animal BA2019. (**A**) Splenomegaly, (**B**) transversal section of the spleen showing splenomegaly, and (**C**) icterus in subcutaneous and mesenteric fat. (**D**) *A. marginale* spherical bodies (arrows). Blood smear stained with May Grunwald–Giemsa, 100× oil immersion. (**E**) Vacuolar hepatic degeneration/necrosis (n) surrounding a centrilobular vessel (vcl). Hematoxylin and eosin staining.

**Figure 3 pathogens-14-00160-f003:**
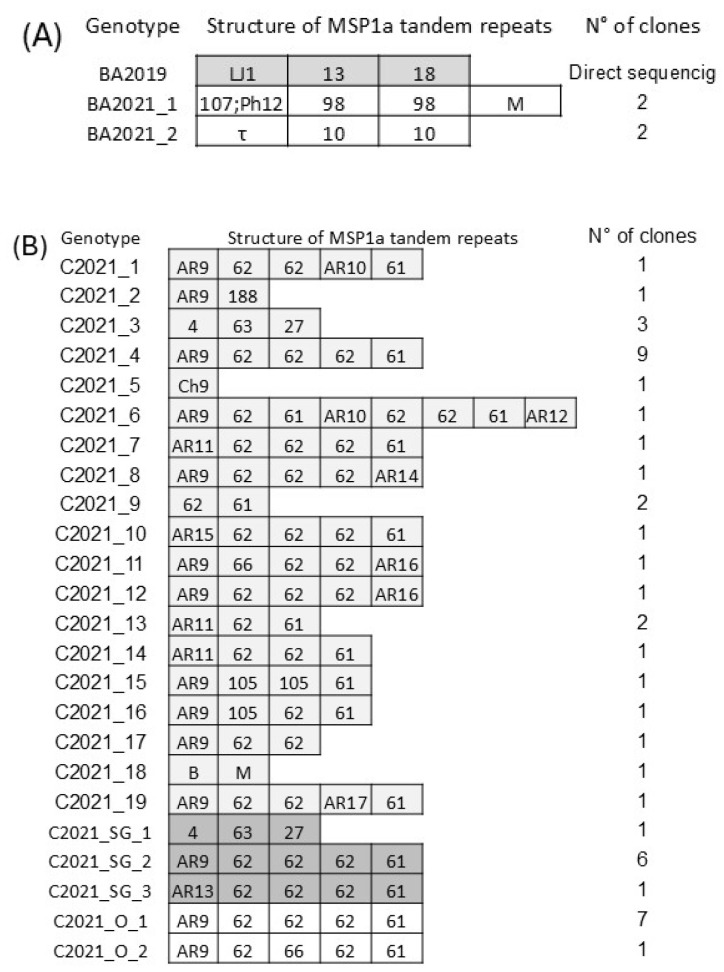
(**A**) *Anaplasma marginale* MSP1a tandem repeats identified in the present study for Buenos Aires (BA2019 and BA2021) samples and the frequency of genotype detection (N° of clones). (**B**) Information on C2021 and tick samples from the same bovine (C2021_SG and C2021_O). The shades of gray indicate the samples from which the genotype shown is derived.

**Figure 4 pathogens-14-00160-f004:**
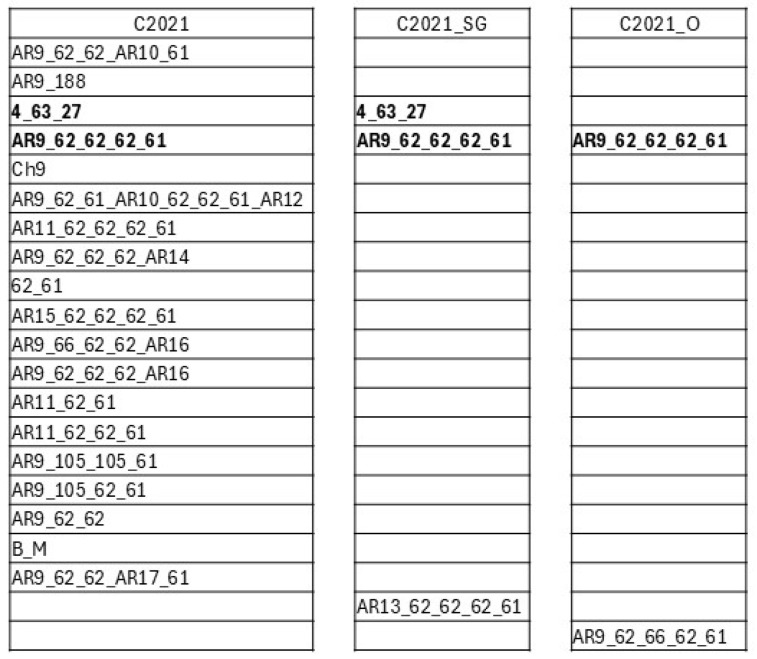
*Anaplasma marginale* MSP1a genotypes found in bovine blood and tick samples (salivary glands and ovaries) from the symptomatic case C2021. Common genotypes between samples are highlighted in bold.

**Figure 5 pathogens-14-00160-f005:**
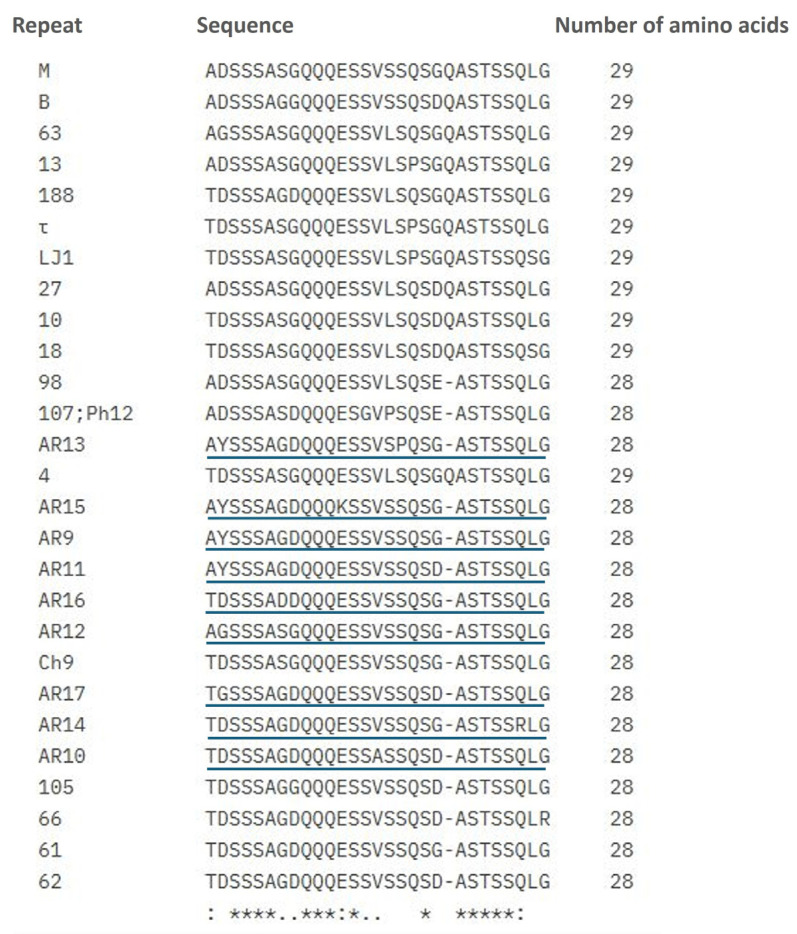
Alignment of *A. marginale* MSP1a amino acid repeat sequences found in this study. Novel identified repeats are underlined. According to consensus symbols, fully conserved residues (*), conserved substitutions (:) and semi-conserved substitutions (.) are indicated.

**Table 1 pathogens-14-00160-t001:** *Anaplasma marginale* molecular diagnosis confirmed the clinical condition of the infected cows through end-point and quantitative PCR.

Sample	Clinical Condition	*msp1β* End-Point PCR	*msp1β* Real-Time PCR
BA2019	symptomatic	Positive	7 × 10^6^
BA2021	asymptomatic	Positive	8 × 10^2^
C2021	symptomatic	positive	1 × 10^7^

## Data Availability

All data generated or analyzed during this study are included in this published article. The datasets associated with the *A. marginale* genotypes generated were submitted to the GenBank database at https://www.ncbi.nlm.nih.gov/gene (accessed on 27 December 2024) and received the following accession numbers: PQ821006-PQ821010 and PQ873055-PQ873102.

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
