# Peer review of "Rhipicephalus microplus and Its Impact on Anaplasma marginale Multistrain Infections in Contrasting Epidemiological Contexts"

_pathogens, 2025, doi:10.3390/pathogens14020160_

Round 1
Reviewer 1 Report
Comments and Suggestions for Authors
The article proposed by Perez et al. focuses on the interesting topic of bovine anaplasmosis among farm animals in Argentina in 2019-2021. Manuscript “Rhipicephalus microplus as a driver of Anaplasma marginale multistrain infections in two different epidemiological settings” described genetic diversity of A. marginale strains isolated from bovine hosts and tick vectors. Article is good written and have clear conclusion about genotyping and ecology Anaplasma genus bacterium. However, not all information were clear described and as Reviewer I have few question and recommendation to improve this manuscript.
General note
The authors of the publication as one of mains topic given the difference in prevalence and genetic diversity of A. marginale in animal hosts and vectors, which is caused by the fact that the vector of this bacterium occurs in one area and not in the other. However, as the authors themselves write, the animals examined in the areas of the non-endemic region for Rhipicephalus microplus, in fact come from the region that the authors consider endemic (transport of cattle after purchase). Additionally, the border between the endemic and non-endemic region is conventional, there is no geographical barrier there (mountain range, large river, large human settlements), so it is not really possible to state with 100% certainty that the region considered non-endemic is completely free from Rhipicephalus microplus. As a reviewer I advise to reconsidered problem, because presented manuscript have a high value in molecular and genotyping analysis of Anaplasma marginale.
Materials and Methods
Please be consist add full reference for used kits, reagents and molecular tools - Name, Company, Country
Results
L223 The numbers given are not sequence numbers. When entering this data into the GenBank database, it is not possible to retrieve data related to this article. Please provide correct sequence numbers.
Discussion
L226 instead “that shapes” please write of
L227 form/in
L238 Please add information about that Cordoba according to Authors is referred as endemic region for Rhipicephalus microplus ticks.
Author Response
We are grateful to the Editor and the Reviewers for his/her attentive reading of our manuscript and helpful insight. In the revised version of the manuscript we modified extensively the approach of the work, introduced the suggested corrections and rewrote some passages. In addition, the full text was reviewed for the writing.
Please find below the detailed responses to the reviewers’ comments and find the corrections in the revised manuscript.
Reviewer 1
General note
The authors of the publication as one of mains topic given the difference in prevalence and genetic diversity of A. marginale in animal hosts and vectors, which is caused by the fact that the vector of this bacterium occurs in one area and not in the other. However, as the authors themselves write, the animals examined in the areas of the non-endemic region for Rhipicephalus microplus, in fact come from the region that the authors consider endemic (transport of cattle after purchase). Additionally, the border between the endemic and non-endemic region is conventional, there is no geographical barrier there (mountain range, large river, large human settlements), so it is not really possible to state with 100% certainty that the region considered non-endemic is completely free from Rhipicephalus microplus. As a reviewer I advise to reconsidered problem, because presented manuscript have a high value in molecular and genotyping analysis of Anaplasma marginale.
We really appreciate the reviewer comments and feedback to improve our manuscript. Regarding R. microplus distribution in Argentina, in the revised version of the manuscript we are adding information from Nava et al. (2024). These authors describe the most updated R. microplus distribution in our country. Based on this information Buenos Aires province remains as a tick-free area including a great area at the north of it. Even though there is no geographical barrier limiting both areas (tick endemic and tick.free), the weather conditions are limiting for this tick species development. Rhipicephalus microplus is not able to complete its biological cycle under Buenos Aires province conditions (Nava et al., 2022; Nava et. al, 2024)
Nava S, Gamietea IJ, Morel N, Guglielmone AA, Estrada-Peña A. Assessment of habitat suitability for the cattle tick Rhipicephalus (Boophilus) microplus in temperate areas. Res Vet Sci. 2022 Dec 5;150:10-21. doi: 10.1016/j.rvsc.2022.04.020. Epub 2022 Jul 2. PMID: 35803002.
Nava S, María NM, Ortega F, María M, Rossner V, Torrents J, et al. EPIDEMIOLOGÍA Y CONTROL DE LA GARRAPATA COMÚN DEL BOVINO RHIPICEPHALUS (BOOPHILUS) MICROPLUS EN ARGENTINA. 2024. 1a ed - Córdoba: EDUCC -Editorial de la Universidad Católica de Córdoba, 2024.Libro digital, PDF - (Producción Bovina. Sanidad Animal ; 1) ISBN 978-987-626-561-4
Materials and Methods
Please be consist add full reference for used kits, reagents and molecular tools - Name, Company, Country
Thanks for the comment, we added the missing information
Results
L223 The numbers given are not sequence numbers. When entering this data into the GenBank database, it is not possible to retrieve data related to this article. Please provide correct sequence numbers.
The datasets associated with A. marginale genotypes in our study were submitted to the GenBank database on 27 december 2024, and accession numbers were assigned to them (PQ821006- PQ821010 and PQ873055- PQ873102). Is possible that the GenBank database may experience a delay before users can access the sequences.
Discussion
L226 instead “that shapes” please write of
L227 form/in
Suggested spelling corrections were made.
L238 Please add information about that Cordoba according to Authors is referred as endemic region for Rhipicephalus microplus ticks.
As mentioned before, in the revised version of the manuscript we are adding information from Nava et al. (2024) regarding R. microplus updated distribution in Argentina. According to this information, in Córdoba province there are tick-free areas (center and south of the province, bordering Buenos Aires province) and areas where R. microplus is present (north of the province).

Reviewer 2 Report
Comments and Suggestions for Authors
Methodology: The Materials and Methods section needs significant improvement. Essential details, such as sample recruitment criteria, statistical analysis methods, and experimental procedures, are either missing or insufficiently described. This makes it difficult to assess the study’s reproducibility and validity.
Sample Size: A larger sample size is crucial for drawing meaningful and generalizable conclusions. The current sample size weakens the statistical power of the study, and this limitation should be explicitly acknowledged in the discussion section.
Results Presentation: The results section requires better organization and clarity. Consider using well-labeled figures, tables, and concise text to guide readers through your findings more effectively. Ensure that statistical outcomes are adequately reported with p-values, confidence intervals, or effect sizes.
Conclusions: The conclusions should be revised to align more closely with the study’s data and limitations. Overgeneralization or extrapolation weakens the credibility of the work. Clearly stating the limitations will enhance the manuscript’s transparency.
Please also check the file attached

Comments on the Quality of English Language
The manuscript would benefit from a thorough review for grammar, syntax, and structure. This will help improve the readability and impact of your study. Consider seeking assistance from a colleague or professional editing service.
Author Response
We are grateful to the Editor and the Reviewers for his/her attentive reading of our manuscript and helpful insight. In the revised version of the manuscript we modified extensively the approach of the work, introduced the suggested corrections and rewrote some passages. In addition, the full text was reviewed for the writing.
Please find below the detailed responses to the reviewers’ comments and find the corrections in the revised manuscript.
Reviewer 2
We really appreciate the reviewer comments and suggestions to improve our manuscript.
Methodology: The Materials and Methods section needs significant improvement. Essential details, such as sample recruitment criteria, statistical analysis methods, and experimental procedures, are either missing or insufficiently described. This makes it difficult to assess the study’s reproducibility and validity.
The primary relevance of the present study lies in the epidemiological impact of the observed findings. Our recruitment criteria was the detection of clinical signs compatible with anaplasmosis and the further confirmation based on molecular diagnosis. While the small sample size prevents a formal statistical analysis, the results nonetheless are relevant since no previous studies have analyzed the genotypic diversity of A. marginale in tick-free areas. Regarding R. microplus endemic regions, our findings in the bovine C2021 are in accordance with previous genotyping studies (Esquerra et al., 2014; Hove et al., 2018; Primo et al., 2021; de la Fourniere et al., 2023).
Sample Size: A larger sample size is crucial for drawing meaningful and generalizable conclusions. The current sample size weakens the statistical power of the study, and this limitation should be explicitly acknowledged in the discussion section.
We are very grateful to the reviewer for highlighting this point. We are not able to increase the sample size of the study because it is an opportunistic sampling. Our study was limited to analyze two clinical cases at different epidemiological conditions. This fact is probably a limitation of our work, and for that reason we added a paragraph in the Discussion section to state it.
“The reduced sample size analyzed in this study is a limitation, and further research with a larger number of samples will contribute significantly to a better understanding of the anaplasmosis epidemiology.”
Results Presentation: The results section requires better organization and clarity. Consider using well-labeled figures, tables, and concise text to guide readers through your findings more effectively. Ensure that statistical outcomes are adequately reported with p-values, confidence intervals, or effect sizes.
In the revised version of the manuscript we added data in the results section and we also modified figures according to it. Regarding statistical outcomes, as we mentioned above, we are showing results from two case reports. We did not make a sample design and the results that we are showing are not adequate for statistical analysis.
Conclusions: The conclusions should be revised to align more closely with the study’s data and limitations. Overgeneralization or extrapolation weakens the credibility of the work. Clearly stating the limitations will enhance the manuscript’s transparency.
Thanks for this comment, we added a paragraph to the discussion section and also added a Conclusions section to the revised version in order to highlight the fact that our results are limited by the sample size.
Conclusions
In the present work we studied A. marginale complex infections in naturally infected bovines from two areas in Argentina. The presence of R. microplus seems to shape a highly diversity profile in the bovine host from the tick endemic area, while bovines in the tick-free area remain carrying a low diversity genotype profile. For the best of our knowledge this is the first analysis of A. marginale complex infections in a tick free area. Further studies involving a larger number of samples will help to improve the knowledge regarding A. marginale complex infections both in tick endemic and tick-free areas.

Reviewer 3 Report
Comments and Suggestions for Authors
Perez et al. investigated the transmission and persistence of Anaplasma marginale strains to bovine hosts in the presence and absence of active Rhipicephalus microplus infestation. Continued tick transmission in areas of Rh. microplus infestation and superinfection resulted in genotype diversity in infected bovine hosts.
line 21: "...the symptomatic bovine holded a highly diversity infection..."
line 65: we aimed to analyze A. marginale strains among tree infected bovine hosts..."
line 67: "In each scenery..."
line 88: The spleen was sampled and conserved (stored?) at -20 degrees C..."
line 107: change to "DNA from bovine blood samples were extracted..."
lines 241-246: Are the genotypic variants BA2021_1 and BA2021_2 novel genotypes? Are these variants associated with bacterial resistance or slow bacterial clearance to oxytetracycline (OTC)? What was the dosing regimen used for oxytetracycline treatment in BA2021 host? Provide clarification in the introduction or discussion sections regarding slow clearing infections following treatment and known genotypes.
Show DNA sequence alignment of msp1alpha tandem repeats for the variants (Figure 2A).
lines 276-277: Explain "These new genotypes inoculated will modify the A. marginale population in the bovine host."
lines 278-885: Explain the role of contaminated medical instuments. Are the authors indicating that the bovine host BA2021 was infected iatrogenically with A. marginale through contaminated instruments or through transmission by tabanid flies, but not by Rh. microplus ticks from the original location of Cordoba?
line 291: "...role in manteining complex infections in the bovine host."
References:
Check reference list format and italicize all genus and species names.
Comments on the Quality of English Language
The entire manuscript needs to be checked for English.
Author Response
We are grateful to the Editor and the Reviewers for his/her attentive reading of our manuscript and helpful insight. In the revised version of the manuscript we modified extensively the approach of the work, introduced the suggested corrections and rewrote some passages. In addition, the full text was reviewed for the writing.
Please find below the detailed responses to the reviewers’ comments and find the corrections in the revised manuscript.
Reviewer 3
Perez et al. investigated the transmission and persistence of Anaplasma marginale strains to bovine hosts in the presence and absence of active Rhipicephalus microplus infestation. Continued tick transmission in areas of Rh. microplus infestation and superinfection resulted in genotype diversity in infected bovine hosts.
line 21: "...the symptomatic bovine holded a highly diversity infection..."
We have modified the text as suggested
line 65: we aimed to analyze A. marginale strains among tree infected bovine hosts..."
We have modified the text as suggested
line 67: "In each scenery..."
We have modified the text as suggested
line 88: The spleen was sampled and conserved (stored?) at -20 degrees C..."
We have modified the text as suggested
line 107: change to "DNA from bovine blood samples were extracted..."
We have modified the text as suggested
lines 241-246: Are the genotypic variants BA2021_1 and BA2021_2 novel genotypes? Are these variants associated with bacterial resistance or slow bacterial clearance to oxytetracycline (OTC)? What was the dosing regimen used for oxytetracycline treatment in BA2021 host? Provide clarification in the introduction or discussion sections regarding slow clearing infections following treatment and known genotypes.
To the best of our knowledge, genotype BA2021_1 (107;Ph12_98_98_M) has not been previously reported. Genotype BA2021_2 (τ_10_10) has been recently reported by our group in bovine and tick samples from Corrientes province (de la Fourniere, 2023). There is no evidence available of these variants regarding antibiotic resistance to Oxytetracycline.
The treatment applied was based on three doses of 20 mg/kg Oxytetracycline 7 days apart.
In the revised version of the manuscript we added a paragraph in the Discussion section in which we mention the previous report of oxytetracycline resistant variants. The following paragraph has been added:
“In a previous report studying A. marginale resistance to long-acting oxytetracycline in bovines, the researchers concluded that some genotypes might be resistant to the drug, thus leading to treatment failure probably due to genotype sensitivity variations, among other factors (15). While the current study did not test the resistance of A. marginale variants to oxytetracycline, it is possible that those genotypes detected in bovines from Buenos Aires after treatment may have exhibited higher resistance profiles. Even if such selection occurred, no further superinfection events were possible since the tick vector is not present in this area (10).”
Show DNA sequence alignment of msp1alpha tandem repeats for the variants (Figure 2A).
As suggested by the reviewer, we added the aminoacid sequence alignment as Figure 5
lines 276-277: Explain "These new genotypes inoculated will modify the A. marginale population in the bovine host."
We have rewritten this sentence for better readability. Now you can read: “These new genotypes, transmitted by ticks to bovines, contribute to increasing the A. marginale population diversity in this vertebrate host.”
lines 278-885: Explain the role of contaminated medical instuments. Are the authors indicating that the bovine host BA2021 was infected iatrogenically with A. marginale through contaminated instruments or through transmission by tabanid flies, but not by Rh. microplus ticks from the original location of Cordoba?
We do not think that bovine BA2021 has been infected in Buenos Aires province, on the contrary, we hypothesize that the herd has been infected in the original location at Córdoba province but due to the natural resistance of calves, the animals did not show clinical signs. We have rewritten this paragraph for a better understanding. Now in the manuscript you can read:
“The animals from the Buenos Aires herd were originally from Córdoba, a province where R. microplus is present in some areas, primary in the north (10). These animals were around eight to twelve months old when moved to Buenos Aires and, at that moment, were not parasitized by ticks nor showed any clinical signs of disease (both are requirements for moving cattle from a tick endemic area to a tick-free area in Argentina). During the anaplasmosis outbreak (BA2019), no iatrogenic practices could be associated with the disease occurrence in the preceding weeks, and all the herd must have been infected when moved from Córdoba. However, as young cattle are resistant to the disease, animals did not show clinical signs until they were older (7). Since R. microplus is absent in Buenos Aires, no further tick-transmission events occurred, and the reduced A. marginale populations in the bovine host were modulated by pathogen-host interactions (29). This explains why asymptomatic bovines were A. marginale positive and harbored a low number of genotypes two years later, in 2021. Regarding A. marginale epidemiology in tick-free areas, we should mention that other haematophagous insects, such as tabanids, have also been associated with mechanical transmission (11)”.
line 291: "...role in manteining complex infections in the bovine host."
We apologize for the typing mistake, we corrected the word “maintaining”
References: Check reference list format and italicize all genus and species names.
We have modified the references as suggested

Round 2
Reviewer 2 Report
Comments and Suggestions for Authors
Line91: Figure 2 only shows gross lesions and that too not all fross observations as described in the figure legend.
Line 95-96: Blood samples were confirmed positive by what method, please describe this.
Figure 5: Please highlight the sequences from this study aligned with published sequences from GenBank. This figure does not show all 11 sequences (AR9-AR19), please show the alignment of all the sequences generated from this study.
Author Response
Line91: Figure 2 only shows gross lesions and that too not all fross observations as described in the figure legend.
Thanks for this comment. We have modified Figure 2 by adding images of histological lesions and also improved images showing macroscopic lesions. In all the images alterations are correctly indicated.
Line 95-96: Blood samples were confirmed positive by what method, please describe this.
Blood samples were tested for A. marginale by amplifying a fragment of msp1β gene. The detailed information regarding this methodology is described in the ”Pathogen detection” section. Now we have added this information also in lines 95-96. In the new version of the manuscript you can read: “One of the 15/20 PCR positive samples was selected for A. marginale characterization (bovine BA2021).”
Figure 5: Please highlight the sequences from this study aligned with published sequences from GenBank. This figure does not show all 11 sequences (AR9-AR19), please show the alignment of all the sequences generated from this study.
Even though all the sequences aligned in Figure 5 were detected in the present study, we underlined the ones that were identified for the first time in our analysis. Regarding these sequences we erroneously wrote in the manuscript that they were called AR9 through AR19, since they are just 9 sequences (from AR9 to AR17). We also made the corresponding corrections in the new revised version of the manuscript. We apologize for this mistake and we really appreciate the reviewer’s correction.

Reviewer 3 Report
Comments and Suggestions for Authors
The authors have responded to all queries.
Minor comments:
Lines 298-302: Rephrase for clarity.
Line 341: "Futer studies involving larger sample..."
Figure 5: Add one to two sentences providing details of what the aligned amino acid sequences show.
Comments on the Quality of English Language
The manuscript needs to be checked for English grammar.
Author Response
Lines 298-302: Rephrase for clarity.
We rewrote the paragraph for clarity. Now you can read:
“The degree of diversity and the genotype profile found in both ticks and bovines depends on specific host-pathogen and vector-pathogen interactions. Among these factors we can mention the microbiome of both bovines and ticks, the immune systems of mammals and arthropods, and the antigenic surface proteins that interact with those immune systems among other factors (24,29).”
Line 341: "Futer studies involving larger sample..."
We apologize for the typing error. We made the correction to “Further studies…”
Figure 5: Add one to two sentences providing details of what the aligned amino acid sequences show.
Thanks to the reviewer for this suggestion. We have added this information to the figure legend. Now you can read:
”Figure 5: Alignment of A. marginale MSP1a amino acid repeat sequences found in this study. Novel identified repeats are underlined. According to consensus symbols, fully conserved residues (*), conserved substitutions (:) and semi-conserved substitutions (.) are indicated.”

Round 3
Reviewer 2 Report
Comments and Suggestions for Authors
Thanks for addressing my comments.